# Effect of Zilpaterol Hydrochloride on Performance and Meat Quality in Finishing Lambs

Jorge Adalberto Cayetano-De-Jesus [1], Rolando Rojo-Rubio [2], Alicia Grajales-Lagunes [3], Leonel Avendaño-Reyes [4], Ulises Macias-Cruz [4], Veronica Gonzalez-del-Prado [2], Agustin Olmedo-Juárez [5], Alfonso Chay-Canul [6], José Alejandro Roque-Jiménez [1] and Héctor Aarón Lee-Rangel [1,*]

[1] Facultad de Agronomía y Veterinaria, Universidad Autónoma de San Luis Potosí, SanLuis Potosi 78000, Mexico; iaz.jorge08@alumnos.uaslp.edu.mx (J.A.C.-D.-J.); alejandro.roque@uaslp.mx (J.A.R.-J.)

[2] Centro Universitario UAEM Temascaltepec, Universidad Autónoma del Estado de México, Temascaltepec 513000, Mexico; dr_rojo70@yahoo.com.mx (R.R.-R.); adal_cay@hotmail.com (V.G.-d.-P.)

[3] Facultad de Ciencias Quimicas, Universidad Autónoma de San Luis Potosí, San Luis Potosi 78000, Mexico; grajales@uaslp.mx

[4] Instituto de Ciencias Agrícolas, Universidad Autónoma de Baja California, Mexicali 21705, B.C., Mexico; lar62@uabc.edu.mx (L.A.-R.); umacias@uabc.edu.mx (U.M.-C.)

[5] Centro Nacional de Investigación Disciplinaria en Parasitología Veterinaria, Progreso 97320, Mexico; aolmedoj@gmail.com

[6] División Académica de Ciencias Agropecuarias, Universidad Juárez Autónoma de Tabasco, Villahermosa 86280, Mexico; aljuch@hotmail.com

* Correspondence: hector.lee@uaslp.mx; Tel.: +52-444-852-4056

**Abstract:** Twenty-four Dorper x Pelibuey lambs were housed in individual pens during a 31-d feeding period and assigned to four treatments ($n = 6$) under a randomized complete block design with different daily doses of zilpaterol hydrochloride (ZH): 0 (control), 0.1, 0.2, and 0.3 mg/kg BW to determine the effects of ZH supplementation on productive performance, meat quality, and wholesale cut yields. Final BW ($p = 0.02$) and average daily gain (ADG, $p = 0.04$) were greater in lambs supplemented with 0.2 mg/kg BW. Supplemental ZH tended to improve dry matter intake (DMI, $p = 0.008$) and ADG:DMI ratio ($p = 0.078$). Wholesale cut yields were not affected by ZH supplementation. Percentage of head was greater ($p = 0.04$) in lambs treated with ZH. The ZH supplementation did not affect carcass characteristics. However, longissimus thoracis et lumborum (LTL) presented a linear trend ($p = 0.08$) of increasing with ZH supplementation. Percentage of blood presented a trend ($p = 0.051$) of decreasing with ZH supplementation. Also, liver decreased in size ($p < 0.05$) for treatments where ZH was included. Values of luminosity decreased ($p < 0.02$) when ZH dosage increased. The value of protein Lowry was greater, with 0.3 mg kg$^{-1}$ ($p = 0.04$). Cathepsin B + L was greater in the lambs from the control treatment ($p = 0.05$). In conclusion, using a daily ZH dosage of 0.2 mg per kg of BW produced the best productive performance, carcass characteristics, and some changes in the meat of hair-breed lambs.

**Keywords:** zilpaterol hydrochloride; meat quality; hair breeds; lamb performance

## 1. Introduction

During the last decade, the number of hair breeds of lambs has increased in several Latin American countries because of their ease of management and resistance to parasites [1]. However, lambs from those breeds have a lower growth rate, carcass yield, and meat quality compared with lambs from wool breeds [2]. The livestock and meat industry is constantly looking for alternatives to promote fast

and efficient growth of livestock, and improve the carcass yield, i.e., increasing *longissimus thoracis et lumborum* (LTL) muscle area [3] or diminishing the fat content in the carcass [4]. Use of growth promoters, such as β-adrenergic agonists (β-AA), may improve growth and carcass characteristics of lambs from hair breeds. Activation of β-receptors in muscle and fat results in increased lipolysis, decreased lipogenesis, and increased protein accretion, or a combination of all these things [5].

Zilpaterol hydrochloride (ZH) is a β-adrenergic repartition agent that has been shown to enhance carcass leanness, improve growth rate, and decrease feed consumption in cattle [6] and lambs [7]. Although zilpaterol was approved for use in feedlot cattle in Brazil, Canada, South Korea, and Mexico, they have been banned in several places, such as China and the European Union (EU) [8]. In the United States, zilpaterol is not currently used in any feeding system due to some people's concern about the possibility that β agonists pose health risks to humans [9]. Human health concerns based on a lack of sufficient information have led some countries to restrict or ban meat imports with traces of ractopamine [10].

The addition of zilpaterol to the diet of lambs [7] and cattle [6] for 30 or 40 d before slaughter has improved daily weight gain, feed efficiency, and carcass characteristics such as hot carcass weight, carcass yield, and LTL muscle area. However, results regarding the effect of ZH on growth and carcass characteristics in lambs are not reliable because there are few reports on the effects on meat quality of productive performance and carcass characteristics. Thus, the objective of this experiment was to evaluate the response in growth performance, carcass yield, and wholesale cut yield of lambs fed ZH at three levels of supplementation, 0.1, 0.2, and 0.3 mg kg$^{-1}$ BW d$^{-1}$.

## 2. Materials and Methods

### 2.1. Location

The experiment was conducted at the Lamb Metabolic Unit at the Centro Universitario UAEM Temascaltepec of the Autonomous University of Mexico State (CUT-UAEM), located in Estado de México, Mexico (19.03° N and 100.02° W).

### 2.2. Ethics

All animal management procedures were conducted following the set of regulations and standards that are required by the Mexican government for the use of animals for various activities: Federal law for animal use and care, humanitarian care of animals during mobilization of animals (NOM-051-ZOO-1995); Federal law for technical specifications for the care and use of laboratory animals, for livestock farms, farms, centers of production, reproduction and breeding, zoos and exhibition halls, must meet the basic principles of animal welfare (NOM-062-ZOO-1995); Federal law for animal health stipulations and characteristics during transportation of animals (NOM-024-ZOO-1995); and Federal law for humanitarian care and animal protection during slaughter process (NOM-033-ZOO-1995).

### 2.3. Animals

Twenty-four hair cross male lambs (3 months of age; initial BW 32.19 ± 0.69 kg) were adapted to individual pens and basal diet (Table 1) for 10 d before starting the experimental phase. Each pen was equipped with feed trough, automatic waterers, and shades. Also, 1 mL of vitamin ADE (Vigantol; Bayer, México City, Mexico) and 0.5 mL lamb-1 ivermectin against internal and external parasites (Ivermectin; Sanfer Laboratory, México City, Mexico; animal-1) was provided on day 1 of the adaptation period.

**Table 1.** Ingredients and chemical composition of the experimental diets.

| Item | Treatments (mg ZH kg$^{-1}$ Body Weight) | | | |
|---|---|---|---|---|
| | **0.0** | **0.1** | **0.2** | **0.3** |
| Ingredients (%, DMI) | | | | |
| Corn | 20 | 20 | 20 | 20 |
| Cob with leaf | 14 | 14 | 14 | 14 |
| Soybean meal | 8 | 8 | 8 | 8 |
| Sorghum | 18 | 18 | 18 | 18 |
| Alfalfa | 11 | 11 | 11 | 11 |
| Wheat bran | 17 | 17 | 17 | 17 |
| Molasses | 9 | 9 | 9 | 9 |
| Mineral premix [a] | 3 | 3 | 3 | 3 |
| Urea | 1 | 1 | 1 | 1 |
| Zilpaterol hydrochloride dose, mg kg$^{-1}$ BW | 0.0 | 0.1 | 0.2 | 0.3 |
| Chemical composition, % | | | | |
| Dry matter | 87.0 | 87.0 | 87.0 | 87.0 |
| Moisture | 13.0 | 13.0 | 13.0 | 13.0 |
| Crude protein | 15.48 | 15.48 | 15.48 | 15.48 |
| Crude fat | 2.20 | 2.20 | 2.20 | 2.20 |
| Ash | 5.61 | 5.61 | 5.61 | 5.61 |
| aNDF | 37.85 | 37.85 | 37.85 | 37.85 |
| ADF | 14.62 | 14.62 | 14.62 | 14.62 |

ZH: zilpaterol hydrochloride; aNDF: neutral detergent fiber; ADF: acid detergent fiber. [a] Ca 270 g, P 30 g, Mg 7.5 g, Na 65.5 g, Cl 100, K 0.5 g, S 42 mg, lasalocid 2000 mg, Mn 2000 mg, Fe 978 mg, Zn 3000 mg, Se 20 mg, Co 15 mg, vitamin A 35,000 IU, vitamin D 150,000 IU, and vitamin E 150 IU, DMI: dry matter intake

During the first day of the experimental phase, ram lambs were weighed and assigned to one of four experimental treatments (*n* = 6). Treatments were a basal diet with one of four daily dosages of ZH (Grofactor®, Virbac Mexico, Guadalajara, Jalisco, Mexico): 0, 0.1, 0.2, and 0.3 mg kg$^{-1}$ BW per lamb (Table 1). Boluses containing 3 g of sorghum meal, molasses, and the respective doses of ZH were mixed and a bolus per day was manually introduced to the oral cavity of each treated animal and control group before the morning feeding.

The experiment lasted 31 d (17 November to 18 December) and experimental doses of ZH were offered during the first 29 d followed by a 2 d withdrawal period before slaughter. Feed was provided three times, at 07:00, 13:00, and 19:00 in proportions of 30%, 30%, and 40%, respectively. All animals had ad libitum fresh water.

### 2.4. Feed Samples Collection

One sample of feed per week was collected, dried in a forced-air oven at 60 °C until constant weight was reached and stored for analysis. All samples were ground (2 mm screen, Wiley mill Model 4; Thomas Scientific, Swedesboro, NJ, USA) and composited to analyse for dry matter (DM), organic matter (OM), ash, ether extract, and crude protein (CP) according to AOAC International [11] methodology. Organic matter content was estimated by subtracting the ash content from the DM content. Neutral detergent fiber (NDF) and acid detergent fiber (ADF) analyses were carried out according to Van Soest et al. [12] using the filter bag technique (ANKOM200 Fiber Analyzer unit; ANKOM Technology, Fairport, NY, USA) with the addition of sodium sulphite and heat-stable amylase to determine NDF.

### 2.5. Finishing-Period Data Collection

Individual BW was recorded before the morning feeding on days 1 and 29 of the experiment. Body weights were reduced by 4% to adjust for gastrointestinal fill. Also, feed offered and refused were measured daily before the morning feeding. From data collected, average daily gain (ADG),

total weight gain, dry matter intake (DMI), and water intake (WI) were calculated based on the amounts offered and refused. Final body weight (FBW) and ADG:DMI ratio were calculated for the overall period.

## 2.6. Carcass Data and Meat Samples

All lambs were slaughtered immediately after the 31-d feeding period in a commercial abattoir. Diet and water were withdrawn 12 h before the slaughter. The methodology utilized to evaluate carcass and non-carcass components was the same as described by Dávila-Ramirez et al. [13].

After being slaughtered by the method of disgorging, the lamb bodies were bled, skinned, and eviscerated to obtain weights of blood, skin, head, foot, heart, liver, lungs, kidney, rumen, small and large intestines, testicles, kidney-pelvic-heart fat (KPH), and hot carcass weight (HCW). After 24 h chilling at 4 °C, cold carcass weight (CCW), carcass length, thorax depth, leg length and perimeter, LTL (using a dot square grid of 64 mm$^2$), and fat thickness were recorded. Additionally, the pH at 45 min and 24 h postmortem (pH45 and pH24) was measured using a portable digital pH meter equipped with a puncture electrode (Hanna, Model HI 98140, USA) in LTL muscle.

Weights of all non-carcass components were expressed as a percentage of the final BW, with the exception of KPH, which was expressed as a percentage of the HCW. Cooling loss percentage was calculated by the difference between HCW and CCW as percentage of the HCW, whereas the dressing percentage was calculated as (HCW/final BW) × 100.

All cold carcasses were cut to obtain the following wholesale cuts: neck, legs, rack and flap, loin, and forequarter and shoulder, according to the meat standards of Australia [14]. The yield of each cut was calculated by expressing its respective weight as a percentage of the CCW.

## 2.7. Color Determination and Lowry Protein

Color was measured on the surface of LTL exposure resulting from the 12th/13th-rib cut, using the lab color space with a colorimeter (Konica Minolta On Color CM-2500d Online, Osaka, Japan). The configuration was aperture size 8 mm, observer 10°, illuminant D65, blooming time 1.5 s. The color coordinates luminosity (Hunter L* Value), redness (Hunter a* value), and yellowness (Hunter B* value) were measured after 24 h, 5 days, and 10 days postmortem. Six determinations were performed on each sample within a determined area. Protein quantitation followed the method described by Lowry et al. [15].

## 2.8. Cathepsin Activities, Myoglobin Content, and Texture Measurements

Cathepsin B and cathepsion B + L activities and myoglobin concentration were quantified 5 and 10 days postmortem in LTLexposure resulting from the 12th/13th-rib cut following Etherington and Wardale [16], and recently described by Negrete et al. [17]. For texture measurements, raw meat toughness was determined in samples of 1 × 1 × 3 cm, 5 and 10 days postmortem, using a compression test, carried out at room temperature (20 + 2 °C), applying up to 20% strain at a speed of 50 mm/min using an Instrom Universal testing machine, model 3365 (Instrom Corporation, High Wycombe, UK), equipped with a modified compresion cell that prevents transverse elongation of raw meat. Five determinations were obtained from each sample.

## 2.9. Statistical Analysis

The results were analyzed according to a completely randomized design using each lamb as an experimental unit. Initial body weight was used as a covariate to account for any unwanted variation within the treatment group. Data were analyzed using the 'mixed' procedure of SAS [18]. Orthogonal polynomial contrasts were used to verify linear, quadratic, or cubic effects for ZH on feedlot performance, carcass traits, non-carcass components, and wholesale cut yields [19], and the means were compared with the Tukey method. Significance was declared at $p < 0.05$.

## 3. Results

### 3.1. Feedlot Performance

The final BW showed a quadratic response (*p* = 0.02; Table 2), with total gain presenting a linear trend (*p* = 0.04) to increase with the increase of ZH. Dry matter intake was affected quadratically (*p* < 0.05) by ZH in the diet. ADG:DMI ratio presented a linear increase (*p* = 0.01) to ZH dosage where 0.2 mgkg$^{-1}$ BW was the best respond.

**Table 2.** Feedlot performance of hair-breed lambs supplemented with different levels of zilpaterol hydrochloride (ZH) during 29 d.

| Item | ZH (mg kg$^{-1}$ Body Weight) | | | | SEM | $p\leq$ | |
|---|---|---|---|---|---|---|---|
| | 0.0 | 0.1 | 0.2 | 0.3 | | l [1] | q [2] |
| Initial BW, kg | 33.02 | 31.64 | 31.65 | 32.45 | 0.30 | 0.20 | 0.01 |
| Final BW, kg | 38.55 | 36.97 | 38.38 | 38.77 | 0.44 | 0.28 | 0.02 |
| Total gain, kg | 5.52 | 5.33 | 6.72 | 6.32 | 0.40 | 0.04 | 0.79 |
| ADG, kg/d | 0.27 | 0.26 | 0.33 | 0.31 | 0.02 | 0.08 | 0.82 |
| DMI, kg/d | 1.52 | 1.33 | 1.33 | 1.46 | 0.04 | 0.43 | 0.01 |
| ADG:DMI ratio | 0.17 | 0.20 | 0.25 | 0.21 | 0.01 | 0.01 | 0.11 |

SEM = Standard Error of the Mean, ADG = Average Daily Gain; [1] linear effect, [2] quadraic effect.

### 3.2. Carcass Characteristics

Treatment effects on carcass composition are shown in Table 3. Hot carcass weight, cold carcass weight and dressing percentage present a linear trend ((*p* = 0.06; *p* = 0.08; *p* = 0.08 respectively) to increase as ZH increases in the diet. Rump perimeter (*p* = 0.05) showed a linear increase (*p* = 0.05) by ZH effect. Also, the pH after 45 min post slaughter, and 24 h post slaughter present a linear increase (*p* = 0.02; *p* = 0.04 respectively). Temperature of LTL increased (*p* = 0.01, quadratic effect) with ZH supplementation. The LTL area cm$^2$ from lambs supplemented with ZH were different between treatments (*p* ≥ 0.05), the 0.3 mg kg$^{-1}$ BW present the major area.

**Table 3.** Carcass characteristics of hair-breed lambs supplemented with different levels of ZH for 29 d.

| Item | ZH (mg kg$^{-1}$ Body Weight) | | | | SEM | *p*-Value | |
|---|---|---|---|---|---|---|---|
| | 0.0 | 0.1 | 0.2 | 0.3 | | l [1] | q [2] |
| Hot carcass weight, kg | 19.11 | 19.03 | 19.76 | 20.58 | 0.57 | 0.06 | 0.44 |
| Cold carcass weight, kg | 18.51 | 18.46 | 19.23 | 19.81 | 0.55 | 0.08 | 0.58 |
| Dressing, % | 48.87 | 49.69 | 52.56 | 51.81 | 1.41 | 0.08 | 0.58 |
| Cooling loss, % | 3.13 | 3.06 | 2.63 | 3.74 | 0.39 | 0.44 | 0.15 |
| pH of LTL | | | | | | | |
| 45 min | 6.75 | 6.68 | 6.78 | 7.03 | 0.09 | 0.04 | 0.09 |
| 24 h | 6.21 | 6.14 | 6.24 | 6.72 | 0.13 | 0.02 | 0.07 |
| Temperature of LTL | | | | | | | |
| 45 min, °C | 21.03 | 20.08 | 20.35 | 21.25 | 0.28 | 0.49 | 0.01 |
| 24 h, °C | 9.40 | 9.27 | 9.48 | 9.40 | 0.14 | 0.75 | 0.88 |
| Carcass length, cm | 61.33 | 63.00 | 61.83 | 61.33 | 1.27 | 0.84 | 0.41 |
| Leg length, cm | 36.66 | 34.33 | 35.33 | 35.16 | 1.02 | 0.46 | 0.31 |
| Perimeter leg, cm | 40.16 | 39.83 | 41.50 | 41.83 | 0.97 | 0.15 | 0.74 |
| Leg width, cm | 16.66 | 16.50 | 15.50 | 16.83 | 0.53 | 0.84 | 0.18 |
| Thorax depth, cm | 22.50 | 22.33 | 22.66 | 22.50 | 0.79 | 0.93 | 1.00 |
| Thorax width, cm | 21.50 | 22.66 | 21.91 | 22.00 | 0.44 | 0.71 | 0.24 |
| Rump perimeter, cm | 61.41 | 61.41 | 62.08 | 63.50 | 0.75 | 0.05 | 0.36 |
| Rump depth, cm | 20.58 | 20.58 | 20.41 | 20.83 | 0.68 | 0.85 | 0.76 |
| Fat thickness, cm | 0.316 | 0.283 | 0.350 | 0.316 | 0.08 | 0.86 | 1.00 |
| LTL area, cm$^2$ | 13.83 | 14.04 | 13.44 | 15.71 | 0.60 | 0.08 | 0.11 |
| KPH fat, kg | 0.471 | 0.351 | 0.390 | 0.413 | 0.04 | 0.52 | 0.15 |

SEM = Standard Error of the Mean; [1] linear effect, [2] quadratic effect.

### 3.3. Commercial Cuts

There was no difference between treatments for the percentages of commercial cuts from carcass lambs according to Australian meat standards (Table 4).

**Table 4.** Commercial cuts from lambs according to Australian meat standards of hair-breed lambs supplemented with different levels of ZH for 29 d.

| Item (% of Whole Cold Carcass) | ZH (mg kg$^{-1}$ Body Weight) | | | | SEM | *p* Value | |
| --- | --- | --- | --- | --- | --- | --- | --- |
| | 0 | 0.10 | 0.20 | 0.30 | | l [1] | q [2] |
| Neck (%) | 4.50 | 4.50 | 4.43 | 4.41 | 0.23 | 0.76 | 0.97 |
| Legs (%) | 32.61 | 33.21 | 32.48 | 33.40 | 0.67 | 0.60 | 0.82 |
| Rack and flap (%) | 17.85 | 15.30 | 15.45 | 15.73 | 1.08 | 0.22 | 0.21 |
| Loin (%) | 15.21 | 15.28 | 14.60 | 15.63 | 0.51 | 0.81 | 0.36 |
| Forequarter and shoulder (%) | 26.23 | 27.91 | 26.28 | 27.56 | 0.88 | 0.56 | 0.82 |

SEM = Standard Error of the Mean; [1] linear effect, [2] quadraic effect.

### 3.4. Percentages of Non-Carcass Components

Treatment effects on percentages of non-carcass components are shown in Table 5. Only a difference of two linear significance was observed. The percentage of liver was a linear increase (*p* = 0.01). Also, the empty large intestine was greater in the lambs supplemented with 0.1 mg kg$^{-1}$ of ZH (*p* = 0.05).

**Table 5.** Percentage of non-carcass components of hair-breed lambs supplemented with different levels of ZH for 29 d.

| Item | ZH (mg kg$^{-1}$ Body Weight) | | | | SEM | *p* Value | |
| --- | --- | --- | --- | --- | --- | --- | --- |
| | 0.0 | 0.1 | 0.2 | 0.3 | | l [1] | q [2] |
| Expressed as % of final BW | | | | | | | |
| Head (%) | 4.57 | 5.10 | 5.21 | 4.99 | 0.20 | 0.16 | 0.09 |
| Blood (%) | 3.94 | 3.65 | 3.44 | 3.50 | 0.16 | 0.06 | 0.31 |
| Skin (%) | 7.94 | 7.01 | 7.75 | 7.48 | 0.35 | 0.69 | 0.37 |
| Heart (%) | 0.44 | 0.45 | 0.44 | 0.40 | 0.02 | 0.17 | 0.27 |
| Lungs (%) | 2.44 | 2.51 | 2.19 | 2.21 | 0.15 | 0.17 | 0.88 |
| Liver (%) | 2.19 | 1.98 | 1.92 | 1.77 | 0.09 | 0.01 | 0.77 |
| Kidney (%) | 0.30 | 0.34 | 0.28 | 0.28 | 0.02 | 0.46 | 0.43 |
| Full rumen (%) | 12.74 | 13.95 | 13.05 | 12.53 | 0.51 | 0.52 | 0.11 |
| Empty rumen (%) | 4.77 | 4.87 | 4.75 | 4.46 | 0.41 | 0.58 | 0.65 |
| Full small intestine (%) | 2.93 | 3.20 | 2.80 | 3.47 | 0.34 | 0.44 | 0.58 |
| Empty small intestine (%) | 1.95 | 2.16 | 1.87 | 1.85 | 0.13 | 0.32 | 0.40 |
| Full large intestine (%) | 4.21 | 4.40 | 4.26 | 4.01 | 0.17 | 0.37 | 0.24 |
| Empty large intestine (%) | 3.17 | 3.33 | 3.16 | 2.64 | 0.18 | 0.05 | 0.09 |
| Testicles (%) | 1.62 | 1.52 | 1.55 | 1.32 | 0.10 | 0.09 | 0.54 |
| Foot (%) | 2.67 | 2.46 | 2.58 | 2.48 | 0.07 | 0.18 | 0.45 |

SEM = Standard Error of the Mean; [1] linear effect, [2] quadraic effect.

### 3.5. Meat Physicochemical Characteristics

The luminosity (L*; *p* < 0.03) redness (a*; *p* < 0.02), and chroma (C) value (*p* < 0.02) decreased when lambs were supplemented with ZH. Also, five days after slaughter, the values of L* and b* (*p* = 0.04) decreased linearly with ZH. At day 10 after slaughter, there was a linear decrease in values of L* (*p* = 0.08), a* (*p* = 0.06), b* (*p* = 0.04), and C (*p* = 0.01). The value of protein Lowry increased linearly (*p* = 0.04) as the dose of ZH increased. However, for cathepsin B at 5 days and cathepsins B + L at 10

days postmortem, there was a trend ($p = 0.07$; $p = 0.05$) to decrease when the dosage of ZH increased (Table 6).

**Table 6.** Meat physicochemical characteristics of hair-breed lambs supplemented with ZH for 30 days.

| Item | ZH (mg kg$^{-1}$ Body Weight) | | | | SEM | *p* Value | |
|---|---|---|---|---|---|---|---|
| | **0** | **0.1** | **0.2** | **0.3** | | **l** [1] | **q** [2] |
| Color 24 h | | | | | | | |
| L* | 35.34 | 33.21 | 33.05 | 29.77 | 1.58 | 0.03 | 0.72 |
| a* | 15.17 | 13.61 | 14.43 | 10.95 | 0.98 | 0.02 | 0.34 |
| b* | 4.41 | 3.93 | 4.07 | 2.46 | 0.62 | 0.06 | 0.37 |
| C | 15.82 | 14.20 | 15.04 | 11.23 | 1.10 | 0.02 | 0.33 |
| H | 15.84 | 15.37 | 15.01 | 12.48 | 1.50 | 0.14 | 0.50 |
| 5 days postmortem | | | | | | | |
| Myoglobin, mg/g | 4.23 | 4.49 | 4.04 | 4.63 | 0.30 | 0.59 | 0.59 |
| Texture, N/cm$^2$ ‡ | 8.23 | 13.38 | 18.54 | 11.16 | 2.34 | 0.20 | 0.02 |
| L* | 36.41 | 33.82 | 33.86 | 30.93 | 1.64 | 0.04 | 0.92 |
| a* | 9.19 | 9.88 | 9.99 | 8.62 | 0.69 | 0.62 | 0.16 |
| b* | 10.76 | 10.18 | 9.97 | 8.04 | 0.83 | 0.04 | 0.43 |
| C | 14.25 | 14.22 | 14.14 | 11.80 | 0.95 | 0.10 | 0.24 |
| H | 48.97 | 45.10 | 44.72 | 43.06 | 2.16 | 0.07 | 0.76 |
| Protein Lowry, mg/mL | 7.80 | 8.30 | 8.99 | 9.55 | 0.59 | 0.04 | 0.96 |
| Cathepsin B ¥ | 0.11 | 0.08 | 0.07 | 0.07 | 0.01 | 0.07 | 0.45 |
| Cathepsins B + L ¥ | 0.19 | 0.16 | 0.13 | 0.14 | 0.03 | 0.17 | 0.41 |
| 10 days postmortem | | | | | | | |
| Myoglobin, mg/g | 4.65 | 4.16 | 4.35 | 4.57 | 0.30 | 0.97 | 0.25 |
| Texture, N/cm$^2$ ‡ | 10.35 | 12.04 | 11.61 | 7.76 | 0.96 | 0.08 | 0.01 |
| L* | 35.92 | 35.10 | 32.26 | 31.71 | 1.81 | 0.08 | 0.94 |
| a* | 10.23 | 8.27 | 8.46 | 7.90 | 0.74 | 0.06 | 0.36 |
| b* | 11.79 | 10.28 | 9.63 | 8.12 | 1.18 | 0.04 | 0.99 |
| C | 15.72 | 13.36 | 12.99 | 11.34 | 1.09 | 0.01 | 0.75 |
| H | 48.64 | 49.80 | 47.61 | 46.01 | 3.57 | 0.54 | 0.70 |
| Protein Lowry, mg/mL | 8.00 | 8.25 | 8.34 | 9.49 | 0.63 | 0.13 | 0.49 |
| Cathepsin B ¥ | 0.10 | 0.06 | 0.05 | 0.07 | 0.01 | 0.09 | 0.71 |
| Cathepsins B + L ¥ | 0.13 | 0.13 | 0.11 | 0.09 | 0.01 | 0.05 | 0.41 |

[1] Linear effect, [2] Quadratic effect; * Six lambs per treatment combination; ‡ Compression test at 20% of total compression; ¥ Expressed as specific activity in nmol of NMec (amino-methylcoumarin) released per min/mg protein.

## 4. Discussion

Growth response increases rapidly at the onset of β-AA feeding until a plateau is reached, and then there is a linear decline in growth due to either down-regulation or desensitization of the β-adrenergic receptors [20]. However, the duration of the response over time is not constant between the different β-AA compounds and species in literature reports [21]. In feedlot lambs, Aguilera et al. [22] and Pringle et al. [23] reported a significant improvement in ADG during the first 2 weeks after administration of the β-AA ZH (6 ppm) and RH (4 ppm), respectively, with no response in the subsequent weeks. However, Kim et al. [24] found a significant increment in total weight gain with a reduction of feed/gain ratio during the first 6 weeks when 10 ppm of cimaterol was added to the diet, which is consistent with our results.

As expected, growth performance and carcass yield responses to supplementation with β-agonists from fattening cattle [25] and lambs [26] depend on the level of dosage. Always with a threshold use, quadratic and cubic effects were found, which means that the responses are not constant because of the dose increase. There are few specific reports about the optimal dosage level of β-AA zilpaterol in feedlot lambs. Salinas-Chavira et al. [27] evaluated the effects of zilpaterol supplementation in feedlot lambs at dosage of 4.35 and 6.0 mg kg$^{-1}$ of dietary DM. Although compared with controls

zilpaterol increased ADG and feed efficiency, no differences were detected between the two zilpaterol dosage levels. In the present investigation, the addition of ZH obtained better total weight gain in feedlot lambs at a dosage of 20 mg kg$^{-1}$ BW, and DMI and FBW presented positive responses at 10 and 20 mg kg$^{-1}$ BW, respectively. Moody et al. [28] proposed causes of variation in growth performance to β-AA supplementation including species, sex, age, genetics, diet, and dosage level consumed. However, Estrada-Angulo et al. [29] concluded that the most important factors in lambs' performance are diet energy density, age, genetics, and ZH dosage level.

Zilpaterol hydrochloride supplementation did not affect carcass weight or dressing percentage. Other studies on β-agonist supplementation reported increments in carcass weight and dressing percentage [23]. Abney et al. [25] and Moody et al. [28] found that carcass yield responds to dosage level of β-agonist supplementation of feedlot cattle and lambs, which disagrees with our results, because the β-agonist supplementation only increases hot carcass yield. Contrary to our results, Elam et al. [6] found effect of exposure time on LM area in studies with feedlot steers supplementing ZH (8.33 mg kg$^{-1}$ of dietary DM) for 0, 20, 30, or 40 d at the end of the feeding period. The authors classified carcasses by LM area and noted a significant increase in the percentage of carcasses with greatest LM area category when ZH was fed, and those carcasses with the highest LM area category increased linearly as duration of ZH feeding increased. In agreement with our results, Vahedi et al. [30] reported no differences in drip loss, cooling loss, and T° at 24 h postmortem in LM muscle of lambs fed ZH, independently of the feeding program applied. The effect on carcass weight loss could be due to an acute reduction in carcass fat content [7].

A fast fall in meat pH during the first 45 min postmortem is associated with a low capacity for water retention and tenderness, whereas a pH greater than 6.0 after 24 h postmortem is associated with dark, firm, and dry meat. These results suggest that this β-AA negatively affected meat quality of ram lambs due to alterations of the normal decrease in the pH during the first 24 h after slaughter. Similar effects of ZH on muscle pH were reported for beef cattle [1] and ewes [13]. However, effects of ZH on muscle pH are limited and contradictory in lambs [31,32]. Information concerning the effect of ZH on wholesale cut yield in lambs is very limited in finishing lambs. Avendaño et al. [1] reported that wholesale cuts were not affected by ZH, except for neck yield, which decreased (based on HCW) with feeding ZH. The percentage of loin is greater in ewe lambs fed ZH [33]. The variation in response to ZH supplementation among wholesale cut yields obtained from lamb carcasses could depend on the presence and amount of type II muscle fibers located in each cut, given that β2-adrenergic receptors are mainly located in these fibers [34]. However, the reason for the lack of ZH effects herein on cuts with high abundance of type II muscle fiber (legs, shoulders, and loin) is unclear.

Non-carcass components were not affected by ZH, which is consistent with results reported by Macías et al. [35] and Avendaño et al. [1]. Elam et al. [6] hypothesized that the increase in HCW and dressing percentage observed in cattle fed ZH could be due to a shift in mass from non-carcass to carcass tissues, especially from visceral organs, or more substrate repartitioning in carcass than in non-carcass tissues, which is unlikely in this research because no increase in muscle mass was found in our lambs. Effects of ZH on visceral organs has been attributed to the difference in the abundance of β-agonist receptor subtypes in these tissues [5]. Ríos et al. [36] reported that ZH decreased liver weight in 9.5% (g kg$^{-1}$ EBW). Similar to our results, Rivera-Villegas et al. [7] found a reduction in liver tissue in the finishing period in lambs using different sources of ZH. Also, the β-agonist salbutamol decreased viscera mass (stomach complex and liver) in pigs. Inasmuch as an appreciable proportion of energy expenditure can be attributed to maintenance of visceral organs, especially the liver and gastrointestinal tract [37], reductions in visceral organ mass could contribute to the increased energy efficiency observed when dietary β-agonists are fed [7].

One of the most important objectives of our study is to contribute to the few studies on the use of ZH in lambs in finishing periods and its effect on meat physicochemical characteristics. The effect on meat color is consistent with the two other studies that reported a decrease in the values of L*, a*, and b* in meat from hair breed lambs [13,38] and the one study that reported it in beef steers [34]

supplemented with different doses of ZH. It seems ZH may have the capacity to affect meat color. A possible explanation for the reduced lightness in the meat of lambs supplemented with ZH is the reduction in the amount of intramuscular fat (marbling), which is white in lambs finished with totally mixed rations [17]. An economic benefit could be the use of ZH supplementation in optimal integration with animal production and meat purveying systems [7]. Hughes et al. [39] suggested that meat color is important for consumer acceptability, with excessively dark meat often associated with consumer rejection. The appearance of meat to the consumer is vital during the purchasing process, with color considered as the main determinant of consumer purchase [40]. Five days postmortem, we observed a statistical trend for the samples of meat from lambs supplemented with ZH treatments. All the ZH treatments showed the lowest amounts for Cathepsin B. The activity of Cathepsin B remained constant until 10 days postmortem. Indeed, in other species, this enzyme requires more time to act [17]. Our findings suggest that the low activity of Cathepsin B in meat from lambs supplemented with ZH during long-term meat aging (3–21 days) parallels the meat tenderization process, revealing an essential role for ZH in reducing cathepsins as proteolytic enzymes in long-term meat maturation. Furthermore, clear differences were detected in the activity patterns of the different cathepsins (B + L) with the use of ZH. Cathepsins B + L were found by Caballero et al. [41] to be important in the later stages of aging (14–21 days). In contrast, the meat from lambs supplemented with doses 0.2 and 0.3 mg kg$^{-1}$ contributed significantly to stopping the increase of Cathepsins B + L. For the increase in meat protein from lambs supplemented with 0.3 mg kg$^{-1}$, our data are similar to Johnson et al.'s [5], which showed that protein degradation was reduced or unaffected by β-AA administration. Also, Wheeler and Koohmaraie [42] indicated that muscle protein degradation was reduced using ZH compared to untreated controls. Additionally, the activity of a specific inhibitor to the calpains, calpastatin, could be elevated in muscle samples [17,41].

## 5. Conclusions

Zilpaterol hydrochloride supplementation is beneficial to feedlot performance and carcass characteristics of economic importance, such LTL area at dose of 0.3 mg kg$^{-1}$ BW, without affecting the wholesale cut yields in hair breed lambs. Some changes in color, protein, texture and cathepsin activity were found by effect of zilpaterol hydrochloride on the meat.

**Author Contributions:** Conceptualization, R.R.-R., L.A.-R., and H.A.L.-R.; Data curation, J.A.C.-D.-J., R.R.-R., A.C.C., and H.A.L.-R.; Formal analysis, R.R.-R., A.G.-L., and H.A.L.-R.; Funding acquisition, R.R.-R. and H.A.L.-R.; Investigation, J.A.C.-D.-J., R.R.-R., A.G.-L., J.A.R.-J., and H.A.L.-R.; Methodology, J.A.C.-D.-J., R.R.-R., U.M.-C., and H.A.L.-R.; Project administration, H.A.L.-R.; Resources, R.R.-R. and H.A.L.-R.; Software, J.A.C.-D.-J., V.G.-d.-P., and H.A.L.-R.; Supervision, R.R.-R. and H.A.L.-R.; Validation, U.M.-C., A.O.-J. and A.C.-C.; Writing–original draft, J.A.C.-D.-J., R.R.-R. and L.A.-R.; Writing–review & editing, J.A.R.-J. and H.A.L.-R. All authors have read and agreed to the published version of the manuscript.

**Funding:** Authors acknowledge the financial support granted by the SEP within the call for "Thematic Networks of Academic Collaboration 2015" to carry out this investigation. Also, they express our gratitude to undergraduate students from CUT-UAEM.

**Acknowledgments:** The first author acknowledges the National Council for Science and Technology (CONACYT, México) for his doctoral fellowship.

**Conflicts of Interest:** The authors declare no conflict of interest manuscript, or in the decision to publish the results.

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
