# Peer review of "Effect of Zilpaterol Hydrochloride on Performance and Meat Quality in Finishing Lambs"

_agriculture, doi:10.3390/agriculture10060241_

Round 1

Reviewer 1 Report

General Comments

The authors investigated the effect of four treatments with different daily doses of zilpaterol hydrochloride (ZH) on productive performance, meat quality and wholesale cut yields and consider that with a daily ZH dosage of 0.2 mg per kg of BW the best productive performance, carcass characteristics and some favorable changes in meat quality are attained. However, the results are, in general, statistically non-significant for carcass traits, commercial cuts, non carcass componentes and meat quality traits. So, the authors should highlight which are the valuable productive gains or scientific innovation and contributions of this manuscript that justify this recommendation.

The description of materials and methods should be more complete. 

Specific Comments:

Line 31: … “However, Longissimus dorsi area increased (P = 0.01) with ZH addition”.

Please, verify. According table 3 the LD increased, but in the 0.1 and 0.3 ZH dosage (not in the recommended 0.2 ZH dosage) and, even so, statistically is non-significant.

Lines 53-54: The authors should briefly state the reasons why the β-adrenergic agonists was banned in several countries.

Line78: A brief explanation about the animals would be helpful for readers.

The animals are all of the same breed? (describe population, history, or genetic relationships)

How old were the animals in the begining of the trial?

How was the management of the animals prior to the study?

Usually, at which weight or age are the animals slaugthered?

Line 105: Justify with references the reduce of 4% of body weights.

Line 119: Please, specify the local of pH measures.

Line 129: Please, indicate the local of colour and protein measures.

Line 135: Cathepsin activities, myoglobin content and texture measurements were done in the same samples of pH and colour determinations? Please, clarify.

Line 138: replace compresion by compression

Line 143: Please, indicate the software used on statistical analysis.

Line 145-146: Add a reference for this analysis. Explain the importance to verify the quadratic or cubic effect.

Line 147: Add the test used for mean comparisons.

Line 154: Table 2

Explain the significance of differences between 0.0; 0.1; 0.2; 0.3 ZH dosage for ADG:DMI ratio, P<0.05. All the mean values are statistically different?

All the mean values of total gain, kg are statistically different?

Line 159: …a statistical trend for higher…

Line 164: All the mean values of rump perimeter are statistically different? Explain.

Line 199: Table 6: LD area was already presented in table 3,do not repeat.

Add myoglobin and texture units

Explain the significance of differences between 0.0; 0.1; 0.2; 0.3 ZH dosage for L*; C, b*, cathepsin B+ L. All the mean values are statistically different? Explain.

Line 201: Quadratic

Line 248: Add reference or delete …” The percentage of loin is greater in ewe lambs fed ZH.”

Line 250: Improve the sentence.

Line 255-256: …no increase in mass muscle were found in our lambs. This variable was evaluated? Explain.

Line 269: Explain the consumer preferences in colour for lamb meat (heavy carcasses).

Line 294: Please verify LD area, the same as line 31. Explain why rump perimeter is of economic importance.

Author Response

Dear Revisor,

We would like to thank you and the reviewers for the comments on the manuscript entitled: “Effect of Zilpaterol Hydrochloride on Performance and Meat Quality in Finishing Lambs.” By Cayetano-De-Jesus et al., We address all the comments and We considering that doing so we improve the quality of the manuscript. 

We have carefully look at all comments and address them. In this file we rewrote the reviewer or section editor assistant comment (EAC), and our response in under author response (R). In the revised version we highlighted the changes in green.

We are happy to address any other comments or concern.

Regards.

General Comments

The authors investigated the effect of four treatments with different daily doses of zilpaterol hydrochloride (ZH) on productive performance, meat quality and wholesale cut yields and consider that with a daily ZH dosage of 0.2 mg per kg of BW the best productive performance, carcass characteristics and some favorable changes in meat quality are attained. However, the results are, in general, statistically non-significant for carcass traits, commercial cuts, non carcass components and meat quality traits. So, the authors should highlight which are the valuable productive gains or scientific innovation and contributions of this manuscript that justify this recommendation.

The description of materials and methods should be more complete.

Specific Comments:

EAC Line 31: … “However, Longissimus dorsi area increased (P = 0.01) with ZH addition”.

Please, verify. According table 3 the LD increased, but in the 0.1 and 0.3 ZH dosage (not in the recommended 0.2 ZH dosage) and, even so, statistically is non-significant.

R= The text was corrected, a linear trend was observed

EAC: Lines 53-54: The authors should briefly state the reasons why the β-adrenergic agonists was banned in several countries.

R=The text was added “In the United States, zilpaterol currently is not used in any feeding system due to some people are concerned about the possibility that β agonists pose health risks to humans (Centner et al. 2014). Human health concerns based on the lack of sufficient information have led some countries to restrict or ban meat imports with traces of ractopamine (EFSA et al., 2009).”

EAC Line78: A brief explanation about the animals would be helpful for readers.

The animals are all of the same breed? (describe population, history, or genetic relationships)

R=The lams were hair cross bread, without genetic relationship.

How old were the animals in the beginning of the trial?

R= 3 months age

How was the management of the animals prior to the study?

R= The lambs received 1 mL of vitamin ADE (Vigantol; Bayer, México City, Mexico) and 0.5 mL lamb-1 Ivermectin against internal and external parasites (Ivermectin; Sanfer Laboratory, México City, Mexico; animal-1) was provided on day 1 of the adaptation period.

Usually, at which weight or age are the animals slaughtered?

R=In the central region of Mexico, the body weight at which the lambs are slaughtered is from 35 kg. Line 105: Justify with references the reduce of 4% of body weights.

EAC Line 119: Please, specify the local of pH measures.

R= pH was measure on the surface of the posterior Longissimus Muscle exposure resulting from the 12th/13th-rib cut.

EAC Line 129: Please, indicate the local of color and protein measures.

R= The measures of Color and Protein were on the surface of the posterior Longissimus Muscle exposure resulting from the 12th/13th-rib cut.

EAC Line 135: Cathepsin activities, myoglobin content and texture measurements were done in the same samples of pH and color determinations? Please, clarify.

R= All measures and chemical meat analysis were done on Longissimus Muscle exposure resulting from the 12th/13th-rib cut.

EAC Line 138: replace compression by compression

R= The concern was addressed and changed

EAC Line 143: Please, indicate the software used on statistical analysis.

R= Data were analyzed using the ‘mixed’ procedure of SAS [18]

EAC Line 145-146: Add a reference for this analysis. Explain the importance to verify the quadratic or cubic effect.

R= Polynomial analysis was used if we are trying to test the dose effect on Y values. Linear could means if we increase the dose level the Y values will increase, and we can select the best level based on the highest dose. Quadratic means, if we increase the dose level the Y values will be increased until certain dose after that the level of dosage will have a negative effect. In animal science cubic effect was not examined because could not have a biological interpretation.

Steel GDR, Torrie JH, Dickey DA (1997)‘Principles and procedures ofstatistics: a biometrical approach.’3rd edn. (McGraw-Hill, New York,NY)

EAC Line 147: Add the test used for mean comparisons.

R= The means were compared with the Tukey test

EAC Line 154: Table 2

Explain the significance of differences between 0.0; 0.1; 0.2; 0.3 ZH dosage for ADG:DMI ratio, P<0.05. All the mean values are statistically different?

All the mean values of total gain, kg are statistically different?

R= The redaction was modified, now the results are expressed by linear or quadratic responses but no by differences.

EAC Line 159: …a statistical trend for higher…

R= statistical trend to linear increase…

EAC Line 164: All the mean values of rump perimeter are statistically different? Explain.

R= The rump perimeter presents a linear increase (p < 0.05) by effect of ZH supplementation, to major dosage is major the rump perimeter of lambs.

EAC Line 199: Table 6: LD area was already presented in table 3 do not repeat.

R= LD was deleted to table 6.

EAC Add myoglobin and texture units

R= The units for myoglobin, texture and cathepsins are:

Myoglobin, mg/g. Texture, N/cm2, ‡Compression test at 20% of total compression. Cathepsins, expressed as specific activity in nmol of NMec (amino-methylcoumarin) released per min/mg protein. The concern was addressed and changed.

Explain the significance of differences between 0.0; 0.1; 0.2; 0.3 ZH dosage for L*; C, b*, cathepsin B+ L. All the mean values are statistically different? Explain.

R= The longissimus-dorsi of lambs supplemented with 0.3 mg kg-1 showed a statistical trend (P= 0.08) to increase (Table 6). The Luminosity (L*; P < 0.03) redness (a*; P < 0.02) and Chroma (C) value (P < 0.02) decrease when lambs were supplemented with ZH. Also, after five days of slaughter, the values of L* and b* (P = 0.04) decrease linearly with ZH At day 10 after slaughter a linear decreased values of L* (P = 0.08), a* (P=0.06), b* (P = 0.04) and C (P = 0.01) showed a lineal decrease. The value of protein Lowry increase linearly (P = 0.04) as increase the dose of ZH. However, for Cathepsins B at 5 days and cathepsin B+L at 10 days postmortem showed a trend (P = 0.07; P = 0.05) to decrease when ZH increase dosage (Table 6).

EAC Line 201: Quadratic

R= The concern was addressed and changed

EAC Line 248: Add reference or delete …” The percentage of loin is greater in ewe lambs fed ZH.”

R= The reference is:

Dávila-Ramírez, J. L., Macías-Cruz, U., Torrentera-Olivera, N. G., González-Ríos, H., Peña-Ramos, E. A., Soto-Navarro, S. A., & Avendaño-Reyes, L. (2015). Feedlot performance and carcass traits of hairbreed ewe lambs in response to zilpaterol hydrochloride and soybean oil supplementation. Journal of animal science93(6), 3189-3196.

EAC Line 250: Improve the sentence.

R= The sentence was retyped by:

The variation in response to ZH supplementation among wholesale cut yields obtained from lamb carcasses could depend on the presence and amount of type II muscle fibers located in each cut, given that β2-adrenergic receptors are mainly located in these fibers (Mersmann, 1998; Walker et al., 2010). However, the reason for the lack of ZH effects herein on cuts with high abundance of type II muscle fiber (legs, shoulders, and loin) is unclear

EAC Line 255-256: …no increase in mass muscle were found in our lambs. This variable was evaluated? Explain.

R= The variable was not evaluated, the sentence was deleted.

EAC Line 269: Explain the consumer preferences in color for lamb meat (heavy carcasses).

R= Appearance, and particularly color, is one of the major factors affecting perception of product quality and its identification (Delwiche, 2004), thus influencing consumers purchasing behavior (Aberle et al. 2001; Resurreccion, 2003). In fact, according to Singh (2006), initial perception of foods occurs within the first 90 s of observation, and approximately 60 to 90% of the assessment is based on appearance

Delwiche, J. (2004). The impact of perceptual interactions on perceived flavor. Food Quality and preference, 15(2), 137-146.

Aberle, E. D., & Forrest, J. C. (2001). Principles of meat science. Kendall Hunt.

Resurreccion, A. V. A. (2004). Sensory aspects of consumer choices for meat and meat products. Meat Science, 66(1), 11-20.

Singh, S. (2006). Impact of color on marketing. Management decision.

EAC Line 294: Please verify LD area, the same as line 31. Explain why rump perimeter is of economic importance

R=LD area on line 31 was corrected. For rump, also referred to as chump, rump from the backside of the lamb where the top of the leg meets the loin. It´s a plump yet lean cut. The economic importance is the generous layer of meat with fat to keep the meat juicy. In different countries, the value of meat cuts is the amount of fat. Also, lamb rump is the most common on top restaurant menus. However, the implications of meat quality and economical aspects is still a point of discussion among scientist, producers and the meat industry, which is ultimately determined by the price paid to the farmer or sheep keepers for the total and individual yield of the most valuable cuts, and the potential differences in sales price and benefits.

Gallo, Sarita Bonagurio, Arrigoni, Mario de Beni, Lemos, Ana Lúcia da Silva C., Haguiwara, Márcia Mayumi Harada, & Bezerra, Helena Viel Alves. (2019). Influence of lamb finishing system on animal performance and meat quality. Acta Scientiarum. Animal Sciences41, e44742. Epub April 29, 2019.https://doi.org/10.4025/actascianimsci.v41i1.44742

Hersleth M, Næs T, Rødbotten M, Lind V, Monteleone E. Lamb meat--importance of origin and grazing system for Italian and Norwegian consumers. Meat Sci. 2012;90(4):899‐907. doi:10.1016/j.meatsci.2011.11.030

Reviewer 2 Report

Review of the research article: Effect of Zilpaterol Hydrochloride on Performance and Meat Quality in Finishing Lambs

General comments

It is not stated what statistical method was applied, however on the basis of the data arrangement in tables, it can be stated that the model was improper. Some of the tables are chaotic, unclear, and improperly designed, and so is the whole results and discussion section. The English needs a major correction – stylistic errors make the results section very confusing, although it can partially be a result of no significance marked.

Detailed comments

Line 44. Increasing the muscle area has nothing to do with carcass yield – by selection, we can change the shape of muscles, change the yield of some carcass parts, but it does not affect the carcass yield.

Longissimus dorsi is an out-of-date term. For a few years, a proper term is longissimus thoracis et lumborum (LTL).

Line 49 – 53. Is it safe for lamb consumers to use zilpaterol in animal nutrition?

Line 128. Please give the CM-2500d settings, for ex. aperture size, observer, illuminant. Also, provide the blooming time for the analysed material.

Line 137. What trait did you measure exactly? A force, energy? In what units were the measures expressed?

Line 142. What statistical procedure was exactly used? If the measures in the same muscle were repeated in time (24 h, 5 and 10 days post-mortem), these measures should be nested in the muscle, nested in the animal.

Line 160. It is not clear. There are no superscripts indicating differences between the values in the table? The reader cannot guess which values differed on p<0.05 level, and which did not differ statistically.

Line 163. From the given p-values, one can conclude about a numerical but not statistically significant differences.

Lin 187.‘The longissimus-dorsi of lambs supplemented with 0.3 mg kg-1 showed a statistical trend (P=0.08) to increase’ – increase in mass? Increase of which trait? Not clear, and only an example, because there are lots of sentences this type. 

All tables – there are no superscripts indicating differences between the values in the tables. Why? How can you state about an effect of different levels of zilpaterol used in feed on any of the analysed traits?

Table 6. Most probably an improper statistical model was used - if it would be a nested model, this table would be differently expressed, as the effect of time post-mortem on the measures should be included.

  • LD area, cm2 - Already given in table 3, and not a physicochemical trait
  • Two different ways of spelling the term post-mortem/ post-mortem? (in this table, and in the main text also)
  • Texture 20%- Define this measurement - what does it express, what units?

Author Response

Dear Revisor,

We would like to thank you and the reviewers for the comments on the manuscript entitled: “Effect of Zilpaterol Hydrochloride on Performance and Meat Quality in Finishing Lambs.” By Cayetano-De-Jesus et al., We address all the comments and We considering that doing so we improve the quality of the manuscript. 

We have carefully look at all comments and address them. In this file we rewrote the reviewer or section editor assistant comment (EAC), and our response in under author response (R). In the revised version we highlighted the changes in green.

We are happy to address any other comments or concern.

Regards.

REVISOR 2

The English needs a major correction

R= The English was send to professional editing

Detailed comments

EAC:  Line 44. Increasing the muscle area has nothing to do with carcass yield – by selection, we can change the shape of muscles, change the yield of some carcass parts, but it does not affect the carcass yield.

R= The observation is correct, but the main of sheep production is increase the productivity and efficiency of sheep meat production are key factors to enhance the competitiveness of this industry (Montossi et al. 2013). The use of β2-adrenergic agonists (β2-AA) as growth promoters has demonstrated to increase sheep meat production by improving protein deposition in muscle (Domínguez-Vara et al. 2013). The use of such technology in the lamb fattening systems could be a nutritional strategy to improve their competitiveness. Over the last decade, zilpaterol hydrochloride (ZH) has been the most widely studied β2-AA and is officially approved for cattle feeding in various countries. Studies done with ram lambs from hair genotypes have reported up to 43 and 35% more daily weight gain and feed efficiency, respectively, when ZH is supplemented (Ríos-Rincón et al. 2010; LopezCarlos et al. 2011). Carcass characteristics of economic importance (i.e. carcass weight, dressing percentage, Longissimus dorsi area) also increased up to 13% by effect of ZH (LopezCarlos et al. 2011).

EAC: Longissimus dorsi is an out-of-date term. For a few years, a proper term is longissimus thoracis et lumborum (LTL).

R= Term Longissimus dorsi was changed by longissimus thoracis et lumborum (LTL)

Line 49 – 53. Is it safe for lamb consumers to use zilpaterol in animal nutrition?

R= Zilpaterol hydrochloride is a beta-agonists approved by the FDA for use in food animal species in the United States (Dilger, 2015), is only approved for use in cattle and binds to beta-2 receptors (Dilger, 2015). These beta-agonists is also approved for use in other countries, such as Brazil, Canada, South Korea, and Mexico; however, they have been banned in several places, as well, such as China and the European Union (EU) (Dilger, 2015). In the United States, zilpaterol currently is not used in any feeding systems. In a meta-analysis of research data that included more than 50 comparisons for both ractopamine and zilpaterol, dietary supplementation in cattle presented notably increased weight gain, HCW, LM area, and G:F (Lean et al., 2014).

Authority began an investigation to evaluate the compound’s safety because it had not done so before adding it to the list of banned veterinary drugs (Bories et al., 2009). Their investigation considered available information about B-agonist from previous research, including studies examining effects on laboratory animals, dogs, monkeys, pigs, cattle, and humans (Bories et al., 2009). Although no new research was conducted, panelists concluded that the detailed investigation did not provide enough evidence to overturn the ban because it was not clearly stated that the consumption of B-agonist residues by humans was safe (Bories et al., 2009), despite approval by FDA in the United States. Hence, the ban was political, as there was no reason to believe that there was a risk to humans, and therefore a nontariff barrier to trade. Nonetheless, the issue will not dissipate any time soon, and especially as more countries push for zero-tolerance, or other nonscientific protocols, relative to beta-agonist residues

This ban was implemented despite conclusions published in several reports by a Scientific Working Group of 22 notable European scientists that was formed by the Commission of the European Communities (the forerunner to the EU) and led by Prof. G. E. (Eric) Lamming of the United Kingdom, that clearly refuted any human health consequences of using anabolic growth technologies in livestock production.

Bories, G., P. Brantom, J. Brufau de Barberà, A. Chesson, P. S. Cocconcelli, B. Debski, J. Dierick, J. Gropp, I. Halle, C. Hogstrand, et al. 2009. Safety evaluation of ractopamine: ESFA panel on additives and products or substances used in animal feed (FEEDAP). EFSA J. 1041:1–52.

Dilger, A. 2015. Beta-agonists: what are they and why do we use them in livestock production? https://www.meatscience.org/docs/default-source/publications-resources/fact-sheets/beta-agonists---dilger-20158d82e7711b766618a3fcff0000a508da.pdf?sfvrsn=69f481b3_0

Lean, I. J., J. M.  Thompson, and F. R.  Dunshea. 2014. A meta-analysis of zilpaterol and ractopamine effects on feedlot performance, carcass traits and shear strength of meat in cattle. PLoS One. 9:e115904. doi:10.1371/journal.pone.0115904

Johnson, R. 2015. The U.S.-EU Beef Hormone Dispute. Congressional Research Service Report. p. 34. https://fas.org/sgp/crs/row/R40449.pdf.

EAC: Line 128. Please give the CM-2500d settings, for ex. aperture size, observer, illuminant. Also, provide the blooming time for the analyzed material.

R= Aperture size 8mm, observer 10°, illuminant D65, Blooming time 1.5 s

EAC: Line 137. What trait did you measure exactly? A force, energy? In what units were the measures expressed?

We used an instrumental method, the most frequently applied instrumental procedure for assessing meat tenderness is the Warner-Bratzler Shear Force (WBSF) test. This test measures the maximum force (N) as a function of knife movement (mm) and the compression to shear (cut off) a sample of meat. The result of this measurement shows the hardness (toughness) of meat. The term shear refers to sliding of meat parallel to the plane of contact, with the applied force tangential to the segment. Nevertheless, this word is commonly used in food technology to attribute any cutting action which splits a product into two fragments.

In the WBSF method, different devices for analysis can be used with a particular head or blade attached to them. These include machines such as Texture Analyzers, Instron devices or other common test devices. Therefore, WBSF is performed either by a unique machine or by some other automatic device with the WBSF blade mounted to it. In the examination, a blade cuts through the meat samples so that shearing is perpendicular to the longitudinal positioning of the muscle fibers. The unit is: N/cm2, Compression test at 20% of total compression

EAC: Line 142. What statistical procedure was exactly used? If the measures in the same muscle were repeated in time (24 h, 5- and 10-days post-mortem), these measures should be nested in the muscle, nested in the animal.

R= The results were analyzed according to a completely randomized design using each lamb as an experimental unit, initial body weight was used as a covariate to account for any unwanted variation within treatment group. Data were analyzed using the ‘mixed’ procedure of SAS [16]. Orthogonal polynomial contrasts were used to verify linear, quadratic or cubic effects for ZH on feedlot performance, carcass traits, non-carcass components and wholesale cut yields (Steel et al 1997), and the means were compared with the Tukey test significance was declared at p < 0.05.

Polynomial analysis was used if we are trying to test the dose effect on Y values. Linear could means if we increase the dose level the Y values will increase, and we can select the best level based on the highest dose. Quadratic means, if we increase the dose level the Y values will be increased until certain dose after that the level of dosage will have a negative effect. In animal science cubic effect was not examined because could not have a biological interpretation.

Line 160. It is not clear. There are no superscripts indicating differences between the values in the table? The reader cannot guess which values differed on p<0.05 level, and which did not differ statistically.

R= The results were present according the statical analysis, text change by:

Treatment effects on carcass composition are shown in Table 3. Hot carcass weight, cold carcass weight and dressing percentage present a linear trend (P≤0.05) to increase as ZH increase in the diet. Rump perimeter (P = 0.05) showed a linear increase (P = 0.05) by ZH effect. Also, the pH after 45 min post slaughter, and 24 h after slaughter present a linear increase (P≤0.05). Temperature of LTL increased (p < 0.05, quadratic effect) with ZH supplementation. The LTL area cm2 from lambs supplemented with ZH had a statistical trend and greatest sizes (Table 3).

Line 163. From the given p-values, one can conclude about a numerical but not statistically significant differences.

R= Result was expressed as a “trend”

Lin 187.‘The longissimus-dorsi of lambs supplemented with 0.3 mg kg-1 showed a statistical trend (P=0.08) to increase’ – increase in mass? Increase of which trait? Not clear, and only an example, because there are lots of sentences this type.

R=The result by longissimus thoracis et lumborum was move to “3.2-. carcass characteristics”, the result was change by “The LTL area cm2 from lambs supplemented with ZH had a trend to linear increase (P = 0.08)

 All tables – there are no superscripts indicating differences between the values in the tables. Why? How can you state about an effect of different levels of zilpaterol used in feed on any of the analysed traits?

R= The way of expressing the results was changed, the correction is adjusted by the statistical analysis used, we used three different doses and analyzed by polynomial arrangement to find the appropriate dose, so we describe the linear and quadratic response so there are no superscripts in the tables

Table 6. Most probably an improper statistical model was used - if it would be a nested model, this table would be differently expressed, as the effect of time post-mortem on the measures should be included.

R= Your observation is correct, we have a great mistake, but it was corrected, the mixed analysis no present effect of the time on the variables analyzed.

  • LD area, cm2 - Already given in table 3, and not a physicochemical trait

R= It was change

  • Two different ways of spelling the term post-mortem/ post-mortem? (in this table, and in the main text also)

R= the term is pot-mortem

  • Texture 20%- Define this measurement - what does it express, what units?

R=Warner Blazer Shear have a blade designs of both the WB and Kramer shear systems. Because differences in crosshead speed can affect shear results, the recommended crosshead speed is 200-250 mm/min. The load cells and full scale load range should be selected suchthat they are between 20% and 80% of range, to meat samples 20% is recommended.

Tenderness, as a specific textural characteristic, can be measured instrumental with compression TPA, penetration, and WB shear values using either triangular or square shear device. The square-cutting device is preferable as it gives results in N/cm2.

Round 2

Reviewer 1 Report

General comments:

The authors answered and corrected some of the issues, however the main problems persist:

There are no superscripts indicating diferences between the values in the Tables.

The auhors use too much the term trend for results statiscally not significant  to justify the results or conclusions and it is not enought in scientific articles.

The conclusion is not supported by the results.

Specific comments:

Line 138: The period of time is not enough for blooming, instead it should be used without blooming.

Lines 161 – 164: There are no superscripts indicating diferences between the values in the Table2. The reader do not know which values differed on P<0.05 and which did not differ statiscally. The same happens in other tables.

For example:

For ADG:DMI ratio (linear effect) the values are all statiscally different?

Line 165: There are no superscripts indicating diferences between the values in the Table 1.

Line 169- 170: Incorrect. P is greater than 0.05  (P>0.05).

Lines 173-174: Incorrect. The trend is not consistent.

Line 301: In which dosage?

Line 302: Again, the results do not support these afirmations, specially for LTL area.

Author Response

General comments:

The authors answered and corrected some of the issues, however the main problems persist:

There are no superscripts indicating diferences between the values in the Tables.

The auhors use too much the term trend for results statiscally not significant to justify the results or conclusions and it is not enought in scientific articles.

The conclusion is not supported by the results.

R= The conclusion was modified according to the results.

Specific comments:

Line 138: The period of time is not enough for blooming, instead it should be used without blooming.

R= Your observation is correct; the blooming period was 15 minutes.

The illuminant, D65 because is closely to daylight.

Lines 161 – 164: There are no superscripts indicating diferences between the values in the Table2. The reader do not know which values differed on P<0.05 and which did not differ statiscally. The same happens in other tables.

R= The supercripts were added in variables with stadistic differences. The differences were analyzed according tukey test.

For example:

For ADG:DMI ratio (linear effect) the values are all statiscally different?

R= Yes, the ZH supplementation present differences respect the control, not only present a linear effect.

Line 165: There are no superscripts indicating diferences between the values in the Table 1.

R= The superscripts were added from table 2 to table 6

Line 169- 170: Incorrect. P is greater than 0.05  (P>0.05).

R= Thank for the observation, it was corrected.

Lines 173-174: Incorrect. The trend is not consistent.

R= The observation is correct; I apologize by the mistake. The results were expressed according the difference between treatment.

The LTL area cm2 from lambs supplemented with ZH were different between treatments (P≥0.05), the 0.3 mg kg-1 BW present the major area respect the other dosages.

Line 301: In which dosage?

R= The best response to LTL area was at dose of 0.3 mg kg-1 BW respect the other treatments

Line 302: Again, the results do not support these afirmations, specially for LTL area.

Zilpaterol hydrochloride supplementation is beneficial to feedlot performance and carcass characteristics of economic importance, such LTL area at dose of 0.3 mg kg-1 BW, without affecting the wholesale cut yields in hair breed lambs. Some changes in color, protein, texture and cathepsin activity were found by effect of zilpaterol hydrochloride on the meat.

Reviewer 2 Report

Dear Authors

The quality of the manuscript has greatly improved. All my doubts considering the statistical analysis and the arrangement of data in tables have been dispelled.

Regards

Author Response

Dear revisor, we are grateful for your observations.

Best Regards
